# Cognitive Remediation in Psychiatric Disorders: State of the Evidence, Future Perspectives, and Some Bold Ideas

**DOI:** 10.3390/brainsci12060683

**Published:** 2022-05-24

**Authors:** Wolfgang Trapp, Andreas Heid, Susanne Röder, Franziska Wimmer, Göran Hajak

**Affiliations:** 1Department of Psychiatry and Psychotherapy, Social Foundation Bamberg, 96049 Bamberg, Germany; andreas.heid@sozialstiftung-bamberg.de (A.H.); susanne.roeder@sozialstiftung-bamberg.de (S.R.); franziska.wimmer@sozialstiftung-bamberg.de (F.W.); goeran.hajak@sozialstiftung-bamberg.de (G.H.); 2Fachhochschule des Mittelstands, Department of Psychology, University of Applied Sciences, 96050 Bamberg, Germany

**Keywords:** cognitive remediation, cognition, neuropsychology, psychiatry

## Abstract

Many people with psychiatric disorders experience impairments in cognition. These deficits have a significant impact on daily functioning and sometimes even on the further course of their disease. Cognitive remediation (CR) is used as an umbrella term for behavioral training interventions to ameliorate these deficits. In most but not all studies, CR has proven effective in improving cognition and enhancing everyday functional outcomes. In this paper, after quickly summarizing the empirical evidence, practical advice to optimize the effects of CR interventions is provided. We advocate that CR interventions should be as fun and motivating as possible, and therapists should at least consider using positively toned emotional stimuli instead of neutral stimuli. Participants should be screened for basic processing deficits, which should be trained before CR of higher-order cognitive domains. CR should stimulate metacognition and utilize natural settings to invoke social cognition. Wherever possible, CR tasks should link to tasks that participants face in their everyday life. Therapists should consider that participants might also benefit from positive side effects on symptomatology. Finally, the CR approach might even be utilized in settings where the treatment of cognitive impairments is not a primary target.

## 1. The Importance of Cognition in Psychiatric Disorders

Currently, many excellent reviews and meta-analyses regarding cognitive remediation (CR) cover either a broader range of different psychiatric disorders [1,2] or yield in-depth discussions for single disease entities, such as schizophrenia [3,4,5], bipolar disorder [6], depression [7], mild cognitive impairment, or dementia/major neurocognitive disorder [8,9]. Therefore, after briefly summarizing the empirical evidence, this review focuses on practical issues for implementing CR as routine therapy in psychiatric wards or outpatient services. We will try to answer the following questions: How should cognitive training be designed to maximize its efficacy? Which additional techniques and components might boost its effects? Could CR be a valuable therapeutic tool, even if the primary goal is not to ameliorate cognitive deficits? 

Whereas deficits in specific cognitive domains are apparent in some psychiatric disorders, such as dementia or Attention-Deficit/Hyperactivity Disorder (ADHD), cognitive dysfunctions are not immediately noticeable, for example, in people with major depressive disorder, schizophrenia, or bipolar disorders. However, do these groups of people nevertheless experience cognitive deficits, and are these deficits relevant for their further course of the illness or their functioning in real life? 

In the last few decades, cognitive deficits have been studied very intensively in people with schizophrenia. Meanwhile, many researchers consider cognitive impairments to be core symptoms of schizophrenia that play a significant role in developing psychotic and affective symptoms and limit social functioning. There is robust evidence regarding stable deficits in various cognitive domains, such as attention, verbal and visual (working-) memory, and executive functions (EF) of about one standard deviation below average [10,11]. These impairments can even be found before first onset [12] and in persons at risk for schizophrenia (children, siblings, and parents of individuals with schizophrenia [13,14]). Robust correlations with social skills, community functioning, social behavior, social problem solving [15], and even the probability of returning to work or school have been identified [16]. These correlations remain significant, even when potential moderator variables are controlled for, such as age, gender, inpatient status, and illness chronicity. Even linkages to the further clinical course of the disease have been reported by some authors [17,18,19]. However, there is significant heterogeneity in cognitive impairment in schizophrenia and other psychoses. Most studies that have attempted to cluster cognitive subtypes in schizophrenia [20,21,22] report three distinct groups. While a smaller group of up to 25% of people with schizophrenia show relatively unimpaired cognitive functioning, a group with intermediate deficits (up to 40%) and a severely impaired group showing global deficits and increased negative symptomatology (up to 60%) could be identified.

Almost the same can be said for bipolar disorders. Even in euthymic phases, many patients present neurocognitive dysfunctions [23,24]. Again, these deficits seem to occur before the onset of the illness [25] and are present in unaffected first-degree relatives of people with bipolar disorder [26]. Although showing a similar profile, these deficits seem to be less pronounced than in schizophrenia [27,28]. Further, for bipolar disorder, adverse effects of cognitive impairment that, in many cases, exceed the influence of residual depressive symptoms on functioning and quality of life could be documented [29,30,31].

Although cognitive deficits have been studied less frequently in major depressive disorder (MDD) than in schizophrenia, there is solid evidence that MDD is also often associated with cognitive deficits, mainly concerning attention, learning and memory, processing speed, and executive function [32,33]. The latter cognitive domain, especially cognitive control, which includes all higher-level processes that enable flexible and adaptative cognition and behavior following current goals, has been discussed as the cause of cognitive deficits in other domains. Moreover, impairment in cognitive control could even contribute to depressive symptoms [34,35]. Similar to schizophrenia and bipolar disorder, these cognitive deficits seem to be present when patients are currently depressed and, to a lesser extent, during euthymic remitted state [23]. It could be shown that cognitive functioning level correlates with the burden and duration of illness [36], social and occupational functioning [37,38], and even the response to psychotherapy or medical treatment [39,40,41].

Furthermore, in ADHD, where the name of the disease already contains one possible cognitive issue, equally pronounced deficits were found in working memory, executive function, and processing speed [42,43].

Besides deficits in attention, working memory, and executive functioning, in substance-use disorders, marked impairments regarding response inhibition and delay discounting were found repeatedly [44]. While deficits in memory and executive functioning seem to be present across different substance types and in multiple types of substance use [45], psychostimulant and alcohol use might affect impulsivity and cognitive flexibility. MDMA use might be associated most strongly with attention and cannabis and methamphetamine with prospective memory (recalling a planned intention in the future) [46]. There are first indications that these cognitive deficits significantly impact the exacerbation and perpetuation of clinical symptoms [47].

Finally, more subtle declines in several cognitive domains might persist, even in other disorders, such as post-traumatic stress disorder (PTSD; mainly attention and working memory and processing speed), obsessive-compulsive disorder (OCD; mainly executive function and memory), or eating disorders (mainly impairments in cognitive flexibility) [48,49,50,51], that may also affect clinical symptomatology [52,53].

## 2. Can We Help?

Unfortunately, neither in schizophrenia nor in bipolar disorder or depression, drugs have succeeded in ameliorating cognitive deficits, apart from effects that an improvement in clinical symptoms can explain [54].

Therefore, it seems reasonable to directly address cognitive deficits via cognitive training. The technical term ’cognitive remediation’ is commonly used in this context, and its most recent definition, given by the cognitive remediation expert group in 2010, reads: “Cognitive remediation is a behavioral training intervention targeting cognitive deficit (attention, memory, executive function, social cognition, or metacognition), using scientific principles of learning, with the ultimate goal of improving functional outcomes. Its effectiveness is enhanced when provided in a context (formal or informal) that provides support and opportunity for extending to everyday functioning” [55].

Meta-analyses that have focused on the effects of CR in schizophrenia [3,4,56] report moderate effect sizes, regarding overall cognition for post-treatment and follow-up assessments. This effect seems to be relatively unaffected by the age of the participants, use of computers, frequency, and duration of the training, as well as by the type of control condition (active or treatment as usual). However, when CR is combined with adjunctive psychiatric rehabilitation (non-pharmacological interventions, such as, for example, social skills training, vocational rehabilitation, or supported employment) and when social functioning instead of cognitive functioning is considered, strategy-based training approaches (e.g., explicit teaching of strategies, such as mnemonics or detailed hints on how best to perform the tasks) appear to be superior to pure ‘drill and practice’ approaches [50,57]. Unfortunately, a meta-analysis of CR trials conducted in first-episode schizophrenia revealed only borderline significant effects of CR on most cognitive domains analyzed and no significant effect on general cognition. However, significant effects on functioning and symptoms could be found. The authors argue that this might be due to a lesser extent of impairments in this subpopulation [58].

For depression, the most recent meta-analysis [59] found significant moderate effects of CR on general cognitive functioning and significant small effects on depressive symptoms and daily functioning. The highest effects were reported for verbal learning and verbal (episodic and working) memory, smaller effects were present for processing speed and attention, and no effects were found for other cognitive domains (EF, verbal fluency, visual learning, and memory). There were no significant effects for any of the variables mentioned above at follow-up. Compared to a previous meta-analysis [7] that found higher effect sizes on each of these domains, more than twice as many studies were included and a less optimistic picture is painted. Generally, it can be stated that the sample sizes in CR studies for people with depression that have been published so far are relatively small (in 14 of 24 studies, the total sample size was 30 or less). Effect sizes are higher when patients with severe depressive symptoms, compared to moderate depressive symptoms, are included and when placebo control conditions that are explicitly designed in order not to improve cognitive functioning instead of waitlist/TAU control conditions are used. Perhaps single-domain training, especially cognitive control training, could be more favorable, as two studies incorporating cognitive control training found effect sizes significantly higher than the average effect size for all studies [60,61].

At the moment, there are few studies with inconclusive results regarding CR in bipolar disorder. While an early study without a control group found promising effects on cognition and residual depressive symptoms [62], the few randomized controlled trials (RCTs) that have been conducted so far paint a mixed picture. The study of Torrent and colleagues [63] found improvements in psychosocial functioning in a large sample of patients with bipolar disorder. Unfortunately, the authors found no effects of CR compared to psychoeducation or TAU on cognitive performance. Two studies from another workgroup found either no effects of CR on cognitive or psychosocial functioning [64] or an isolated effect on a single measure of executive functioning and on subjective cognitive functioning that disappeared at follow-up six months later [65]. Finally, Lewandowski and colleagues [66] reported post-treatment and follow-up (24 weeks after termination of CR) effects for a composite measure of cognition but only significant effects on one of seven cognitive domains for post-treatment (visual memory) and follow-up (processing speed).

For ADHD, recent meta-analyses [67,68] found no significant effects of CR on attention when measured by neurocognitive tests but effects on working memory, even when active control conditions were included in the RCTs. Nevertheless, significant effects on ADHD symptoms of inattention have been demonstrated. However, the effect sizes did not reach significance when blinded ratings were performed. There is evidence that multidomain cognitive training might be superior to single-domain cognitive training [69,70,71], as people with ADHD exhibit deficits in multiple cognitive domains, which are related to different brain regions, and transfer measures, such as academic achievement, might be the result of a composite influence of different cognitive domains. Newer approaches use combined working memory tasks with inhibitory control components, such as the n-back task, and some studies have proven effects on cognitive and symptom outcomes [68,72,73,74].

A recent review [45] identified 32 studies regarding the effects of CR on substance use. Only 12 of these were classified as high-quality studies, and 20 of them had a moderate risk of bias due to lack of assessor blinding, insufficient or no randomization, adherence to treatment, or high or not documented dropout rates. Participants were mainly alcohol users, but some studies also included stimulants, methamphetamine, polysubstance, cannabis, opioids, and ketamine users. These studies yielded mixed results, so there is no clear indication of evidence. Although working memory training is currently the most popular, the best results were obtained when multiple cognitive domains were targeted. Some studies also reported effects on treatment outcomes (relapse rates, improved commitment to therapy).

Further, for anorexia nervosa, the results are currently inconsistent regarding the effects on cognitive flexibility, the only cognitive domain consistently documented as impaired in affected patients [51,75,76]. Recent meta-analyses [77,78] found no significant effects of CR on cognitive flexibility but on self-reported executive functioning behavior (behavioral regulation and metacognition) in young people with anorexia nervosa. Additionally, two recent RCTs with higher sample sizes found no CR effects on BMI change, anorexia nervosa and obsessive-compulsive symptoms, motivation to change, or cognitive flexibility [79,80].

In mild cognitive impairment (MCI), a recent meta-analysis and a recent review have shown moderate effect sizes on cognition that are larger for people with MCI than for unimpaired older adults [8,81]. Such effects have been found for verbal and nonverbal working memory, memory, and learning domains, but to a lesser extent for processing speed and executive functioning—the latter being a key predictor of functional decline [82]. However, an intervention that specifically targeted vision-based processing speed could demonstrate effects on working memory and processing speed [83].

In major neurocognitive disorders, such as dementia, there is evidence that CR approaches cannot be applied in the same way as for other psychiatric disorders and might have to be modified. While early reviews [84] found moderate effect sizes on cognition, the mean sample sizes for the studies included were relatively small, and when only high-quality studies were included, effect sizes dropped to a non-significant value. As these high-quality studies used active control groups, the authors suggested that people with dementia might profit more from general cognitive stimulation than CR approaches. Consequently, subsequent reviews found that interventions involving a wide range of activities, such as discussing past or recent events, listening to or making music, completing small cognitive tasks, or engaging in practical activities, such as tinkering or cooking, have a significant impact on cognitive and social functioning [85].

## 3. How Can We Do It Right?

Despite impressive evidence regarding the effectiveness of CR, there is a considerable amount of variance in effect sizes and a substantial number of findings yielding no effect of CR interventions. Consequently, much thought has been given to identifying success factors of cognitive training. In the remaining text of this manuscript, we present some suggestions that might boost the effects of CR (see also Figure 1).

### 3.1. Identify Patients in Need

One approach could include only patients who experience significant cognitive impairments (usually defined by scores of least one standard deviation below the normative mean in at least one targeted domain)—a strategy utilized by many studies cited above. There is evidence that better results are obtained in participants with larger cognitive deficits before the onset of CR [65,86]. However, while this may be useful to maximize effect sizes in clinical trials, this could exclude patients motivated to participate in clinical CR practice. Already, when CR is delivered individually, and even more, when CR is used in groups, some basic form of social interaction (e.g., feedback about performance on the task, exchange on how difficult the individual exercises are perceived to be, discussions about promising strategies, etc.) will take place. This type of social interaction may be less fear-inducing for participants than more complex real-life social interactions. Regardless of improved cognitive performance, performance in a computer-based task is improved simply by completing it multiple times. Therefore, the potential beneficial effects of CR on social competence, self-efficacy, and sense of mastery might be valuable in ameliorating non-cognitive symptoms of mental illness because the motivation to attend other therapies might increase (see also the explanations in the following sections of this chapter).

### 3.2. Check for Basic Processing Deficits

Recently, there has been some debate whether a more basic perceptual training of lower-level auditory and visual processing or a more elaborate training of higher-order cognitive functions should be performed. The results of a recent high-quality trial, which compared training of executive functioning (executive control impairments might be a core feature of schizophrenia [87,88]) against perceptual training, found effects on neurocognition, functional competence, and community functioning, twelve weeks after treatment (but not post-treatment, see below) for the executive functioning but not for the perceptual training group [89]. In explaining these results, the authors point out that enhanced executive functioning might lead to greater engagement with more cognitively stimulating environments. However, the perceptual training reported in this study might have been underdosed. Furthermore, by prior detection and amelioration of early auditory processing dysfunction, affected persons might profit even more from CR [90]. Thus, checking whether there are basic processing deficits in the first step, and if such deficits exit, using a combination of both approaches, could be the most effective strategy (see also [91] for a thorough discussion of this topic).

### 3.3. Consider Thinking Outside the Box

#### 3.3.1. Social Cognition Training in CR Settings

In Chapters 3.6 and 3.8, we will discuss that social cognition is an essential aspect of CR and should be included in CR interventions. Further, we will highlight that every intervention helping to transfer newly gained cognitive abilities into everyday life is very welcome. For this reason, many current CR concepts contain elements beyond completing paper and pencil or computerized cognitive tasks.

One simple step away from merely completing single-player tasks could be to administer CR, at least partly, in groups. First results indicate that even basic interaction between participants during CR interventions, such as noting each other’s scores [92], might boost its effects. However, in our opinion, one could go even further. One reason why the combination of psychiatric rehabilitation and CR is superior to CR alone (see Section 2) is the fact that attending rehabilitation interventions, such as vocational rehabilitation or supported employment, is hardly possible without having a social exchange with other participants or without using the cognitive skills acquired by the CR intervention. For this reason, the software we use in our current CR trials is different from the CR tasks used in former studies [19,93,94]. Two new types of task are incorporated. The first group consists of exercises that can only be solved as a group together and when the group members communicate (see [95] for examples of such tasks included in a preliminary earlier version). The other tasks can be played either in single-player or group mode. In the latter mode, either the entire group tries to achieve a target score, formed by the sum of all individual scores (which, of course, are saved individually and evaluated against participant’s high scores), or the tasks are played in a competitive mode, where two groups play against each other, and the group achieving the highest total score wins. In group mode, it is displayed during runtime whether the current performance predicts that the target score is reached or whether the participant group’s sum score is below or above the other group’s sum score. This setting was chosen to facilitate discussions among the group members. For example, participants could talk about which strategies might be helpful and exchange tactics they have applied successfully in the past. As explained in Section 3.2 and Section 3.6, this might benefit the participants’ interpersonal skills and symptom levels.

#### 3.3.2. Combination of Physical Exercise and CR

Meanwhile, there is some evidence related to the effectiveness of physical activity on cognitive performance for schizophrenia [96,97], depression [98], Alzheimer’s disease/mild cognitive impairment [99], and ADHD [68]. One possible explanation for these results is that physical activity seems to stimulate the release of neurotrophic factors, such as Brain-Derived Neurotrophic Factor (BDNF) [100], which promotes synaptogenesis, neurogenesis, and angiogenesis [101,102]. The physical effort also leads to increased oxygen and glycose metabolism [103]. Thus, it is possible that combined physical activity training and CR could increase effects, as the physiological changes described above might facilitate the neuroplastic effects of cognitive engagement during CR.

The most straightforward approach is to apply both interventions separately [104,105]. Interventions that perform CR exercises during aerobic physical activity, such as ergometer riding [106], would take this idea one step further. However, for example, it would even be possible to realize tasks that utilize GPS tracking and require the participants to walk/run a certain physical distance or a predefined number of steps to complete the (gamified) cognitive exercise.

Recently, a positive effect of combined physical and cognitive training via so-called “exergames”, gamified cognitive tasks that have to be played via physical activity (e.g., movement on a step training platform), has been documented in people with major neurocognitive disorder and healthy older adults [107,108].

#### 3.3.3. CR as a Tool to Improve Clinical Symptoms

Finally, cognitive tasks could be presented in settings where the primary target is not to ameliorate cognitive deficits but to ameliorate symptoms. In Section 2, we reported that cognitive training can improve depressive symptoms in MDD or negative symptoms in schizophrenia. However, improvements in non-cognitive symptoms were viewed as pleasing and beneficial side effects in this context, as the primary goal was to improve cognitive deficits. Nevertheless, it would be conceivable to use CR tasks when the primary aim is to improve symptoms. Currently, our workgroup is evaluating a virtual reality (VR) app for people with claustrophobia (see Figure 2). Virtual reality exposure therapy has proven effective in treating anxiety disorders, with effect sizes comparable to in-vivo exposure therapy [109]. The novel aspect of our approach is that participants have to perform a gamified attention task while being in a simulated claustrophobic environment. Spheres embedded in the surrounding walls must be found and touched following particular rules (a button on the controller must be pressed when target spheres, but not when other spheres, are touched). By completing the task, participants are forced to respond differently from their learned reaction triggered by the fearful situation (e.g., moving around quickly, approaching and touching the walls, constantly scanning the environment, etc.). This intervention might trigger counterconditioning [110], adding to the extinction effect usually achieved by plain (VR) exposure.

### 3.4. Have Fun!

One significant key variable suggested by many authors [57,111,112] is participants’ motivation, which, in turn, is influenced by many other variables. First of all, the therapists running the CR intervention can contribute to patients’ motivation by explaining to them why cognitive training is a helpful tool, by giving information about the cognitive domain(s) trained by each task and by providing positive feedback regarding performance improvements or by praising the process of training (attending the sessions, staying focused, etc. [55]). To accomplish this, therapists should have some experience concerning the tasks administered and have some background knowledge about neurocognition.

Participants’ insufficient motivation can become a problem when the substantial dropout rates in the most recent high-quality RCTs for schizophrenia of up to 70%, from randomization to follow-up, are considered [89,113]. These observations point to the enormous importance of keeping patients engaged in CR interventions.

Naturally, features of the software itself also play an essential role. Ideally, the fun factor of an attractive computer game is combined with the face validity of a cognitive test—or, put in more modern words, its tasks are gamified. Gamification, defined as the use of game design elements in non-game contexts, is a feature that has proven to be highly engaging, on the one hand, and boosts participants’ motivation, on the other hand [114]. However, to gamify a cognitive task, it is not enough to simply implement elaborate graphics and animation. It is also essential to have compelling plots for each cognitive exercise. For example, participants might have more fun preventing hungry snails from eating lettuce they have planted by remembering where the snails have been hiding than watching butterflies and completing tasks, such as “point to the butterfly that landed on the yellow flower”.

Scaffolding [115], as another helpful feature to keep attendees motivated, means putting them to their limit but not beyond (often, a cut-off of 80% correct performance before progression to a higher level of difficulty is recommended). In this way, an optimal level of difficulty can be found, and participants can be prevented from either becoming frustrated or bored.

Since noting an increase in performance is motivating, everything that indicates progress, such as obtaining a new personal high score, reaching a new game level, etc., is welcome.

### 3.5. Consider Utilizing Hot Cognition

Most research regarding CR utilizes an information-processing view of cognition. In this theoretical framework, input delivered by the body’s sensory systems is transformed by the human brain into internal representations, which can then be manipulated. Therefore, sensory and motor systems are considered as remote components, distinct from cognition, merely acting as interfaces. The model of working memory by Baddeley [116,117] is paradigmatic for such an approach, such as in a personal computer, a central executive works, such as a computer processor, which retrieves information from permanent storage units, modifies them, and temporarily stores the results into auditory or visual random access memory (RAM) modules.

Cognitive models of depression-like Beck’s schema theory [118,119] or associative network models [120], however, have pointed out that in human beings, information is not always processed as impartial as in a computer. Instead, these models predict that dysfunctional cognitive structures and cognitive biases affect perception, attention, memory, and reasoning and play a significant role in developing, identifying, and maintaining affective disorders. Indeed, it could be shown for several cognitive domains that the performance of people with MDD differs from healthy controls in so-called “hot cognition” tasks that use emotionally relevant stimuli instead of stimuli without any emotional influence [121]. For example, stimuli related to depression are better recalled by people with MDD than neutral and happy stimuli [122,123], and there is evidence that this kind of altered “hot cognition” might not only be a state marker of depression but might also function as a vulnerability factor [124,125].

In this context, a lot of interest has also been given to attentional biases, as attention plays an important role in other cognitive domains and emotion processing and regulation [126]. Attentional biases have been studied extensively for social anxiety disorders, where a bias toward threatening stimuli [127], and to a lesser extent, for depressive disorders, where a bias towards negatively evaluated stimuli could be found [126]. This means that people with social anxiety disorders/depression tend to look toward threatening/sad stimuli, and they do so without even realizing it.

Consequently, whether these biases could be modified and whether this modification could reduce clinical symptoms has been investigated. Typically, a bias shift in the desired direction, i.e., away from threatening or sad stimuli, is “rewarded”. For example, participants would have to quickly react to neutral cues (for example, a simple dot, “coins”, or whatever in a more gamified approach), which appear less often in areas with sad or threatening stimuli. Indeed, small effects on the primary symptoms [128] and a CBT-boosting effect in younger participants [129] for social anxiety disorder could be demonstrated.

Although results are more disappointing in depressive disorders concerning clinical effects [130,131], a recent result from a high-quality RCT using more elaborate hardware gives hope that similar effects in depression might also be obtained [132].

To our knowledge, no studies that have combined classical CR approaches and bias modification interventions have been published yet. Given that tasks to modify cognitive biases could be easily transferred into the patients’ everyday lives, this is surprising. Using mobile versions, higher training volumes could also be achieved for bias modification tasks. A “hot–cognition variant” would also be conceivable for an everyday episodic memory training task: In the first step, events, appointments, or shopping items could be evaluated for their positive and negative valence. The task could then be designed so that content with positive valence must be remembered before neutral and negatively rated content is queried.

In this context, it should be mentioned that, although people with psychiatric disorders often routinely receive pharmacotherapy, little attention has been drawn to the beneficial or adverse impact of medication on the effects of CR. For example, in MDD, improvement in symptoms caused by antidepressive medication should indirectly improve cognition because cognition correlates with the severity of depression [133]. Although current antipsychotic drugs have limited effects on cognitive deficits in schizophrenia, a recent RCT could show that the form of administration (oral vs. long-acting injectable) determined whether CR was superior to an active control condition or not [134].

### 3.6. Include Social Aspects of Cognition

Social cognition includes all mental operations regarding perception, interpretation, and understanding of social information [135]. Deficits in social cognition, which correlate with neurocognitive impairments, are also linked to functional outcomes [136] and could be more closely related to community functioning [137]. Social cognitive performance appears to be inversely associated with the severity of depression [138] and has been shown to predict interpersonal skills in individuals with schizophrenia [139]. Although conventional CR can improve social cognition [4], current approaches have tried to remediate social cognition directly.

Typical tasks used in such interventions (see e.g., [140]) are exercises regarding recognition of emotions (e.g., correct identification of facial expressions or assigning pairs of faces with the same emotional expressions), perceiving social cues (following of eye movements to quickly identify target stimuli), remembering social facts (e.g., hobbies of fictional people), and Theory of Mind (understanding emotions of persons in social situations). Indeed, there is first promising evidence for the effectiveness of such interventions on social cognitive performance and social functioning [141,142]. However, a recent RCT of the same workgroup found no superior effects of combined social cognitive and perceptual training compared to perceptual training alone, on symptoms, quality of life, or social functioning [113].

Nevertheless, we would advocate strengthening the inherent social components of CR in a targeted manner to increase its effects. First, in schizophrenia, as in many other psychiatric conditions discussed above, social anxiety symptoms that might moderate the link between social competence and performance [143,144] are common. Using group comparisons of participants with high and low levels of social competence and social performance, the authors found that social anxiety was correlated with the levels of social performance but not with the levels of social competence. Even in the group with low social competence and high social performance, better global functioning and quality-of-life scores were achieved compared to those with high social competence and low social performance. These intriguing results are comprehensible; independent of the skill level required, relevant social situations often may be too complex and too emotionally overwhelming for the participants of CR interventions. Therefore, administering CR in groups might gradually expose participants to interactive social environments while providing a structured way of exploring and training social skills and social performance.

Second, some concepts, such as the Neuropsychological Educational Approach to Rehabilitation (NEAR [145]), besides trying to relate tasks to everyday problem-solving or vocational challenges, explicitly use social interaction and peer support provided in group settings. Thus, approaches, such as NEAR, might also improve social cognition. Further, when CR and rehabilitation are combined, the social interaction between the participants might partly be responsible for the superior effects concerning social functioning. This is because a rehabilitation intervention includes having social exchanges with other persons, which then might improve social cognition.

Summing this up, a combination of basic social cognition training targeting social skills, such as recognizing facial expressions and prosodic fluctuations or identifying gaze directions, as described above, should be combined with approaches where these skills can be applied later on during CR interventions that utilize group settings.

### 3.7. Encourage Thinking about Thinking

Recent contributions distinguish between two CR approaches: ‘drill and practice’ and ‘strategy training’. Through the plain repetition of cognitive exercises, therapists hope that cognitive achievement improves, through implicit learning, a cognitive domain that seems to be unimpaired in most participants (e.g., in people with schizophrenia [146] or with milder/major cognitive disorder [147]). ‘Strategy training’, on the other hand, refers to the explicit teaching of strategies either by a therapist or by the software itself during cognitive exercises. It is expected that these strategies might help participants to cope with cognitive demands in everyday life. The current understanding is that—at least concerning neurocognition—both approaches are equally efficient [4,56]. However, there is weak empirical evidence that strategy-based approaches may boost the effects of other, for example, vocational rehabilitation programs, even when CR does not improve cognition immediately [148,149]. One reason for the equivalence of ‘drill and practice’ and ‘strategy training’ might be that, even in ‘drill and practice’ approaches, patients might generate strategies by themselves, especially when the tasks’ appearance promotes this. For example, one task in the training software used in our workgroup’s interventions [19,93,94] is playing sounds (e.g., a roaring lion, a plane flying by, etc.) that must be memorized. Listening to a familiar sound usually generates a corresponding mental picture of the item. Thus, elaborative cognitive strategies that are known to facilitate recall are evoked automatically [150]. Generally, self-generated strategies might be better than strategies suggested by clinicians [151]. Therefore, as mentioned above, the tasks themselves and the training set should give room for the development of such strategies. Different tasks for the same cognitive domain should be available to aid in the generalization of these strategies, which the participant should apply implicitly at the end; while conscious recourse to strategies, for example, through self-verbalization, is still desirable in the beginning, participants should ideally use these strategies automatically later on.

A process that has considerable potential to promote the development and use of strategies and may mediate the effects of neurocognition on functional outcomes is metacognition [152], a term coined in its original conceptualization as “thinking about thinking” [153]. Two metacognitive subprocesses seem to be necessary to transfer cognitive improvements into functional improvements. Metacognitive knowledge means knowledge about cognitive processes per se, whereas metacognitive regulation handles monitoring and planning cognitive processes [154]. Metacognitive knowledge includes the awareness of cognitive strengths and weaknesses so that problems are not underestimated, and cognitive resources can be utilized effectively to meet the specific task. On the other hand, metacognitive regulation helps coordinate cognitive operations and decide which strategy or plan should be followed, or if a plan should be changed. Thus, metacognition can help acquire and consciously remember strategies across different learning environments during CR. While some skills learned within CR may immediately be transferable to everyday tasks, in most cases, metacognition is needed to recognize the demands of such an everyday task and apply suitable strategies and appropriate resources.

Promising effects of metacognitive approaches using concepts, such as the web-based application CIRCuiTS, on the amount of use of metacognition [155] and community functioning could have been demonstrated lately. However, the authors could only demonstrate post-treatment and not follow-up effects [86]. Nevertheless, these results make it worthwhile to include metacognitive exercises in future CR approaches specifically.

### 3.8. Help to Build Bridges

As already stated at the beginning of this manuscript, the ultimate goal of CR is not only to improve cognitive performance but to improve functioning in real life. Therefore, improvements in cognition are suboptimal if these improvements do not generalize to enhancements in social and occupational functioning or a better quality of life. Thus, training strategies that force a transfer of the skills and strategies achieved during CR to challenges in everyday life would be desirable.

Much thought has been given to how such a bridging into the participants’ real lives can be accomplished. One recently suggested strategy is simply to encourage participants to phrase their recovery or functional goals, which they consider to be related to employment, social skills, and everyday functioning [5]. Ideally, these goals are translated into ecologically valid targets regarding the mastery of the CR tasks.

In the NEAR approach [145], described above, regular group discussions are held on managing the problems encountered in everyday life and setting appropriate personal cognitive training goals. Other approaches, such as action-based CR [65,156], further extend this idea by adding practical activities, such as remembering messages or even role plays.

Additionally, both approaches encourage the participants to perform CR exercises at home. In this circumstance, apps for smart devices containing either the same or similar tasks used in the clinical labs’ CR interventions might be the first practical step to reaching out of the laboratory into the participants’ real lives. Creating such software has become much easier now. Modern game engines, such as Unity^®^ [157] or Unreal^®^ [158], offer solutions for multiplatform development while drawing minimal demands on programming skills, so CR experts should be able to directly transform their ideas into CR applications (see also [159] for a more detailed discussion of this topic).

In addition, mobile technologies offer the possibility to utilize cognitive demands arising in everyday life directly. For example, a shopping list created on a smartphone could serve as an episodic memory CR exercise: The users would first have to freely remember the shopping items during their stay in the supermarket. Items already in the shopping cart could then be compared with the shopping list, and memory aids (maybe initially the first letter and later perhaps a semantic description) could be given. Mobile systems could also help with daily structuring. Initially, support regarding appointments could be provided (e.g., by reminders of upcoming appointments that have already been agreed upon or are still to be agreed upon), and in a second step, these appointments could be used for training episodic memory (free or cued recall) and executive functions (planning of travel and preparation times, an optimal sequence of appointments, etc.).

Augmented reality apps that run on participants’ smart devices might further aid in bridging CR into everyday life and in gamifying these everyday-life tasks. In the example above, visual images or capital letters of the items that must be shopped could be presented visually in the shopping cart. Virtual coins could lead to the items that have to be collected, and virtual barriers, which indicate unfavorable paths, could appear—the possibilities seem to be unlimited.

### 3.9. Brace for (Advantageous) Side Effects

Whether hospitalized or forensic patients were included, whether inpatients or outpatients participated in the studies, favorable effects on cognition were shown for all these groups. However, for example, in patients in forensic wards, two studies, including people with schizophrenia, also reported improved functioning and decreased aggression [160,161]. Such effects regarding non-cognitive domains have also been seen for depressive symptom levels in MDD [7] or negative symptoms in schizophrenia [162]. There are various conjectures as to why this is the case. It might be that challenges in everyday life and therapeutic tasks can be mastered better via an improvement in specific cognitive domains, such as working memory and executive functioning. Another hypothesis is that motivation is increased by improving reward sensitivity by CR [163]. Heightened self-esteem and self-confidence due to perceived progress in the CR tasks might elevate motivation to engage in other therapeutic interventions [164]. Finally, CR might increase the connectivity of frontal regulatory brain areas with areas associated with emotion regulation, leading to more effective processing of emotionally relevant information [52].

### 3.10. Take Advantage of the Butterfly Effect

The term butterfly effect has been coined in chaos theory, where it could be shown that in complex systems, containing non-linear relationships, even very subtle differences in the initial states of two nearby identical systems can lead to unpredictable deviations in the development of these systems over time. Thus, stated in extreme terms, the flap of a butterfly’s wing might (but, of course, more likely, not) cause a cyclone or, stated in more moderate terms, sometimes, small changes might have a significant future impact [165].

As stated above, several studies have found no immediate but delayed effects of CR. For example, in a recent high-quality trial, effects on neurocognition, functional competence, and community functioning could be shown twelve weeks, but not immediately, after treatment [89]. Such results are surprising, as higher effect sizes immediately after the CR intervention would be expected that decrease over time. Maybe even marginal improvements in cognitive functioning achieved by CR interventions may pay off in the long run by leading to a higher engagement in real life, which might be cognitively stimulating. Combining CR with subsequent rehabilitative interventions that stimulate cognitive and social activities to promote such effects makes sense. In some cases (see above for details), this might explain why a combination of CR with adjunctive psychiatric rehabilitation appears to be superior to pure CR [57].

In practice, it may pay off to give the cognitive deficits an adequate amount of time to heal and ensure that CR does not remain an isolated intervention but is embedded in other psychosocial therapies.

## 4. Conclusions

In the previous pages, we showed that cognitive deficits are present in many psychiatric disorders and that these deficits have a significant impact on the lives of the people affected.

Previously, most data were collected for schizophrenia, where there has been extensive research since the 1990s [166]. However, solid empirical data exist that CR might help ameliorate cognitive deficits and improve social lives and wellbeing for many other disorders, such as major depression, bipolar disorder, ADHD, and mild or major neurocognitive disorders.

Much space has been given to ideas regarding the practical application of CR. In general, we advocate a generous use of CR as a therapeutic tool, not only in its traditional application (improvement in cognitive deficits) but also, for example, when the therapeutic relationship is to be strengthened or when clinical symptoms are to be ameliorated. However, such an extended application requires a respectful relationship with the participants and CR tools that motivate the participants and are fun to use.

More specifically, we have advocated:Choosing a gamified approach that lets participants have as much fun as possible;Combining CR with previous perceptual training if basic processing deficits are present;Including and stimulating metacognition;Using group settings to stimulate social cognition;Letting CR reach out into everyday life as much as possible;Utilizing the effects of hot cognition on information processing;Keeping in mind that the effects of CR interventions might be subtle at the beginning but grow stronger over time;Extending the classic CR paradigm by including cognitive group tasks that require social interaction, combining CR with physical exercise, and considering using CR also in settings where the primary target is to improve clinical symptoms rather than ameliorate cognitive deficits.

Finally, Figure 3 once again summarizes how key features in the CR setting might interact with key outcome variables.

In our future vision, CR will move out of cognitive labs and into environments, such as ‘CR lounges’, that the participants will perceive as more pleasant. Mobile devices, such as tablets or even participants’ smartphones, instead of stationary desktop computers will be used. Connected devices will facilitate administration (e.g., starting, stopping, or monitoring) of tasks that the therapists can give as group or individual exercises. CR will not be limited to psychiatric facilities. Instead, there will be a seamless transition, as participants can continue their training at home, during or after outpatient treatment. If desired and appropriate, blended groups of inpatients, outpatients, and even their relatives or friends that have the option to participate online, will become state-of-the-art CR treatment. In everyday life, cognitive challenges will be utilized for CR using augmented reality and combined with physical activity. CR’s power to ameliorate clinical symptoms will be utilized even in settings where cognitive deficits are not the primary therapeutic target.

At the time of writing this manuscript, the COVID-19 pandemic is causing an enormous global disease burden [167]. There is evidence that some symptoms persist months after the illness, even in cases with a relatively mild course of the disease, with impairments in memory and attention being amongst these symptoms [168,169]. To our knowledge, the effects of CR on long-COVID have not been studied yet. Nevertheless, we would like to see therapists and researchers begin to use their expertise for this patient group and start sharing their experiences and empirical results as soon as possible.

## Figures and Tables

**Figure 1 brainsci-12-00683-f001:**
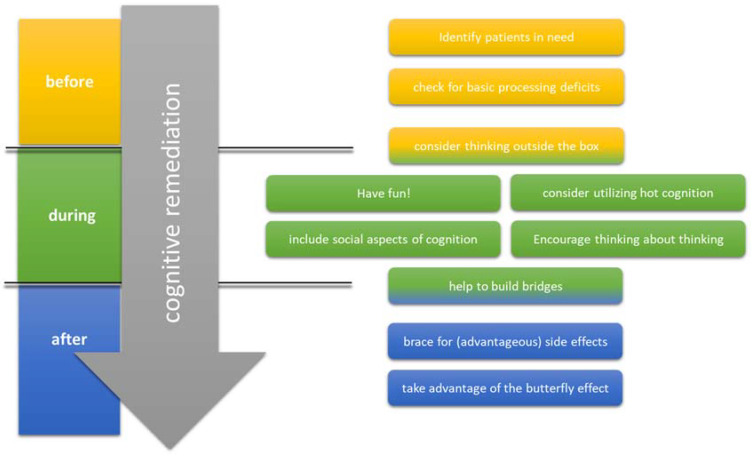
Strategies that could help amplify the effects of CR.

**Figure 2 brainsci-12-00683-f002:**
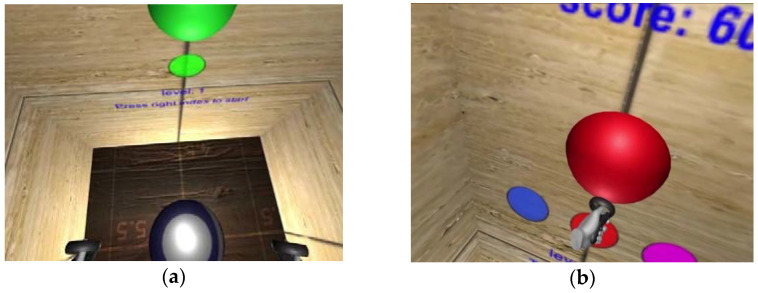
Virtual reality exposure therapy for claustrophobia with an added CR task. (**a**) Before the start of the gamified attention task; (**b**) during the gamified attention task.

**Figure 3 brainsci-12-00683-f003:**
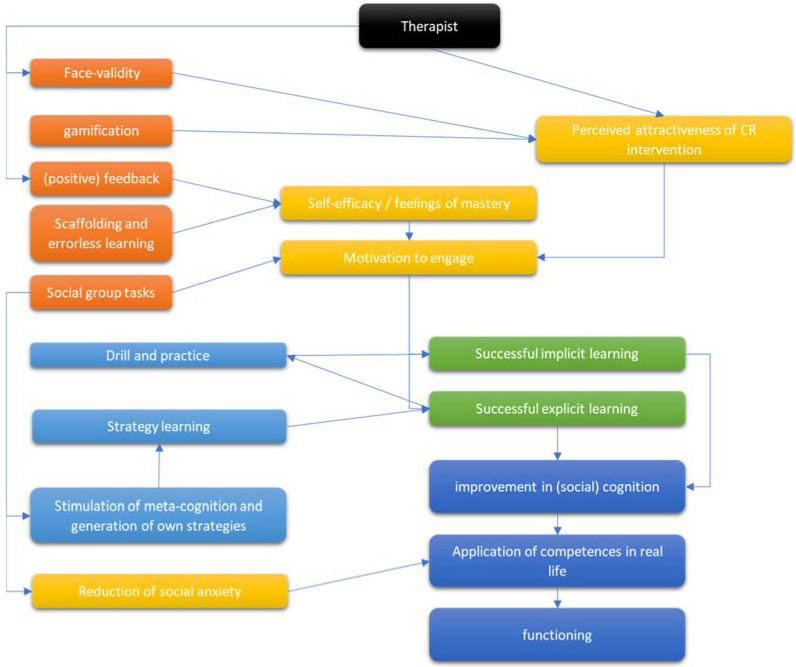
Interaction between key variables in CR settings. Yellow: moderating or mediating non-cognitive variables; orange: features of the CR tasks; light blue: features of the CR intervention; green: types of learning; dark blue: effects of the CR intervention.

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
