# Peer review of "Cognitive Remediation in Psychiatric Disorders: State of the Evidence, Future Perspectives, and Some Bold Ideas"

_brainsci, 2022, doi:10.3390/brainsci12060683_

Round 1
Reviewer 1 Report
In this manuscript, the authors provide a narrative review of the literature of cognitive remediation (CR) as an intervention for various psychiatric and neurocognitive disorders and propose novel approaches to enhance the effects of CR on these various disorders. While the authors make compelling arguments for these points, the manuscript is in need of some major revisions.
Abstract
Line 11: “CR has proven effective not even in improving cognition” implies that CR has does not improve cognition. Consider revising language such as “CR has proven effective not only in improving cognition …”
Line 13: clarify further what the author means by “optimize the output.” This is not a term typically used when speaking about behavioral interventions.
Line 21: need to provide keywords
Introduction
Line 34: avoid using the word “ingredients” when describing an intervention. Consider using intervention-appropriate words such as components.
Line 59: the authors express that cognitive deficits in MDD have not been studied extensively. This is not accurate. There are strong lines of research focused on delineating cognitive function in MDD as well as existing work on the effects of CR, specifically computerized cognitive training, on depressive symptoms. I suggest that the authors revise this paragraph and also add findings related to deficits in cognitive control as this has been the major target for CR in depression.
Sections 1.1 and 1.2: I believe these 2 sections should be combined. The information, presented as is, seems disjointed. It would be better for the flow of the manuscript to combine them, so the author presents the evidence on cognitive deficits among the multiple psychiatric disorders, along with how these deficits affect individuals with these symptoms. They already allude to this merge on the last sentence of the first paragraph (lines 41, 42, 43) where they state whether individuals with psychiatric disorders experience cognitive dysfunction and whether these deficits are relevant.
Line 68: are these findings among individuals with substance use disorder related to alcohol? Where are the strongest findings in the literature pertaining to the type of substance used?
It would be good to describe what the observed cognitive deficits are in anxiety and stress disorders such as anxiety, PTSD, and OCD. Evidence is increasing on these disorders, and recently attention has shifted to understand the effects of CR particularly on anxiety and PTSD symptoms.
Line 111: to strengthen the argument for using CR to complement other interventions, it would be helpful to provide examples of this. What is an example of a strategy-based approach?
Line 120: the authors raise an important point, and that is that the effects of CR on depressive symptoms appear to be variable. They suggest that sample sizes in the studies can be one of the reasons. Addressing whether the difference in types of cognitive remediation interventions (computerized cognitive training vs non-computerized CT) and multidomain vs single domain could contribute to the lower effect sizes reported in the meta-analysis would be an important contribution to the manuscript.
Line 144: ADHD paragraph, addressing why a multidomain approach may be beneficial over a single domain approach to intervene on ADHD would be another important contribution, and may strengthen the case for multidomain CR approaches for all psychiatric disorders.
Line 164: above, the authors mention that there are some declines in cognition in eating disorders, but they don’t specify what are the domains affected. Here, they do. It may be more fitting to move this information above, where they talk about cognitive deficits in eating disorders as it will align better with the flow of the manuscript and specify that most of the work has been done in anorexia nervosa (if this is true). In this section, they can expand on the findings of the meta-analyses as the effects of CR on eating disorders is not a topic of broad discussion and thus it would better educate the reader.
Line 169: please use the appropriate term for MCI which is Mild Cognitive Impairment, not disorder. If adhering to the DSM classification of “Mild Neurocognitive Disorder” then it would help to describe it as such, and state something to the likes of “In Mild Neurocognitive Disorder, or most commonly known as mild cognitive impairment (MCI)…” Also, the authors should be cautious to say that processing speed and executive function are rarely targeted by CR. The ACTIVE trial showed effects of speed of processing training on several cognitive domains and depressive symptoms among older adults. Also, there is published work using computerized cognitive training targeting processing speed on older adults with MCI (for example, see Feng (Vankee) Lin). Additionally, the majority of the work utilizing CR for depression focuses on cognitive control, a component related to executive function. Thus, it would be beneficial if the authors provide an alternative conclusion for these mixed findings that it is specific for patients with MCI.
Line 176: the phrase “in the same way” is repeated twice.
Line 180: I believe the authors meant to say “studies” rather than “stories”
Line 181: correct to say “patients with dementia” rather than “people with dementia patients”
Line 188: This is the first time the authors mention “CT”, I assume they are referring to cognitive training? It would help to spell it out given it is the first time they mention “CT”, or simply use CR to be consistent throughout the manuscript.
Line 234: It would be good for the authors to describe why and how they assume that CR would have beneficial effects on social competence, self-efficacy, and sense of mastery. This is a relatively novel idea and should have some discussion and literature to support it.
Line 280: the paragraph suggesting metacognition as a combinable approach to CR appears to be underdeveloped. It would be helpful for the authors to elaborate on how metacognition can be applied to develop and use strategies during training and how applying metacognition can facilitate transfer of the effects of training on neurocognition to functional outcomes.
Section 3.5: in this section, the authors make a strong case for intervening to improve social cognition. However, the argument falls week for responding to why this is important. As it reads, the argument appears to support targeting social cognition on those with difficulties with social interactions or interpersonal skills, such as individuals with Schizophrenia. Although the authors do mention that social anxiety may be a common thread with other disorders, it is not entirely clear how a combined intervention targeting social cognition and neurocognition can benefit an array of individuals with psychiatric disorders and cognitive deficits. Thus, the authors should expand on this.
Section 3.6: on the section “Help to build bridges” the authors provide excellent concrete examples on the application of cognitive training to everyday functional activities and how these can facilitate better training outcomes.
Section 3.7: regarding conjectures about the effects of CR on non-cognitive domains in MDD and schizophrenia, Quiñones et al. 2020, which the authors already cite, propose that CR may strengthen functional connectivity of functional regulatory areas associated with emotion regulation and thus this may be worth mentioning as another potential explanation for CR effects on clinical symptoms of these disorders.
Section 3.8: it would be helpful for a broader audience for authors to define the concept of “hot cognition.”
Section 3.9.1: the statement that CR often takes place in groups may be an overgeneralization from CR interventions with schizophrenia. This is the same issue as with section 3.5. CR in studies with other patient populations such as patients with depression, PTSD, or MCI is commonly delivered individually, and this is also true for clinicians delivering CR in clinical settings. The authors should state clearly how social cognition exercises benefit individuals with difficulties in social interactions and how this approach could further benefit individuals with other mental health disorders.
Section 3.9.2: the paragraph on physical exercises and CR appears to be inconclusive. It would be helpful if the authors addressed how combined physical exercises and CR may improve cognitive and clinical symptoms. For example, exercise promotes cognitive and emotional brain pathways that are also strengthened by CR, resulting in improved cognitive function and clinical symptoms of psychiatric disorders.
Throughout the manuscript:
It is very important that authors avoid using biased-language. For example, the field of behavioral sciences has moved from using the word “suffer” to other descriptors such as “experience.” Another example in the manuscript that the authors should avoid is “psychiatric patients”; describe individuals as “patients with psychiatric disorders” or “patients with ADHD.”
The flow of section 3 does not follow the flow of Figure 1. Consider structuring this section according to the figure to facilitate the flow or timing of the recommendations.
Authors are encouraged to revise the manuscript for English language grammatical structure.
Author Response
Thank you very much for your thorough review, encouraging words, and many helpful comments. We think that the quality of our manuscript has improved considerably after considering your hints.
Line 11: “CR has proven effective not even in improving cognition” implies that CR has does not improve cognition. Consider revising language such as “CR has proven effective not only in improving cognition …”
Thank you very much, we have changed this sentence accordingly
Line 13: clarify further what the author means by “optimize the output.” This is not a term typically used when speaking about behavioral interventions.
Thank you very much for this hint. We have replaced the word “output” by “effects,” and we hope you agree
Line 21: need to provide keywords
Thank you so much. In the original manuscript version, the keywords were actually present, but somehow they got lost. Please excuse this mishap.
Introduction
Line 34: avoid using the word “ingredients” when describing an intervention. Consider using intervention-appropriate words such as components.
You are right. In line 38, we have replaced the word "ingredients" with "components" as suggested
Line 59: the authors express that cognitive deficits in MDD have not been studied extensively. This is not accurate. There are strong lines of research focused on delineating cognitive function in MDD as well as existing work on the effects of CR, specifically computerized cognitive training, on depressive symptoms. I suggest that the authors revise this paragraph and also add findings related to deficits in cognitive control as this has been the major target for CR in depression.
Please excuse, what we wanted to say is that cognitive deficits have been studied less extensively compared to schizophrenia. Of course, there is solid evidence for depression, too, as we have stated at the end of the sentence.
You are right, that executive functioning is regarded as a key cognitive deficit in MDD, and we have added information regarding this point. However, we did not go into extreme detail in this topic (such as explaining Miyake’s model of EF), as deficits in cognitive control do not seem to be specific for MDD. Some authors, including our own group, think that problems with the proper timing of cognitive control processes are an essential deficit in schizophrenia. We hope this is o.k. for you.
Sections 1.1 and 1.2: I believe these 2 sections should be combined. The information, presented as is, seems disjointed. It would be better for the flow of the manuscript to combine them, so the author presents the evidence on cognitive deficits among the multiple psychiatric disorders, along with how these deficits affect individuals with these symptoms. They already allude to this merge on the last sentence of the first paragraph (lines 41, 42, 43) where they state whether individuals with psychiatric disorders experience cognitive dysfunction and whether these deficits are relevant.
Thank you very much. We agree and have merged the two sections mentioned as suggested.
Line 68: are these findings among individuals with substance use disorder related to alcohol? Where are the strongest findings in the literature pertaining to the type of substance used?
Thank you very much for this remark. No, these are results that were found for different substances. We have added some lines regarding the converging areas across different substances and some notes about the differential effect of different substances on cognition.
It would be good to describe what the observed cognitive deficits are in anxiety and stress disorders such as anxiety, PTSD, and OCD. Evidence is increasing on these disorders, and recently attention has shifted to understand the effects of CR particularly on anxiety and PTSD symptoms.
Good point, thank you very much. We have added this information.
Line 111: to strengthen the argument for using CR to complement other interventions, it would be helpful to provide examples of this. What is an example of a strategy-based approach?
Thank you very much. We have added some explanations.
Line 120: the authors raise an important point, and that is that the effects of CR on depressive symptoms appear to be variable. They suggest that sample sizes in the studies can be one of the reasons. Addressing whether the difference in types of cognitive remediation interventions (computerized cognitive training vs non-computerized CT) and multidomain vs single domain could contribute to the lower effect sizes reported in the meta-analysis would be an important contribution to the manuscript.
This is an interesting point. Unfortunately, all but one of the studies (Twamley et al., 2019) used computerized cognitive training. Maybe, single-domain training, especially cognitive control training, could be more favorable, as two studies incorporating such training found effect sizes that were significantly higher than the average effect size for all studies.
Legemaat et al., 2021 discuss that patients with severe depressive symptoms compared to moderate depressive symptoms and placebo control conditions instead of waitlist/TAU control conditions yield higher effect sizes. We have added both thoughts to the manuscript.
Line 144: ADHD paragraph, addressing why a multidomain approach may be beneficial over a single domain approach to intervene on ADHD would be another important contribution, and may strengthen the case for multidomain CR approaches for all psychiatric disorders.
Thank you very much. We agree and have added some thoughts about this point.
Line 164: above, the authors mention that there are some declines in cognition in eating disorders, but they don’t specify what are the domains affected. Here, they do. It may be more fitting to move this information above, where they talk about cognitive deficits in eating disorders as it will align better with the flow of the manuscript and specify that most of the work has been done in anorexia nervosa (if this is true). In this section, they can expand on the findings of the meta-analyses as the effects of CR on eating disorders is not a topic of broad discussion and thus it would better educate the reader.
Done, thank you very much ?. Recent empirical data from high quality studies is very disappointing, so we have added a few lines that further weaken the general conclusion.
Line 169: please use the appropriate term for MCI which is Mild Cognitive Impairment, not disorder. If adhering to the DSM classification of “Mild Neurocognitive Disorder” then it would help to describe it as such, and state something to the likes of “In Mild Neurocognitive Disorder, or most commonly known as mild cognitive impairment (MCI)…”
Thank you very much, we have corrected that.
Also, the authors should be cautious to say that processing speed and executive function are rarely targeted by CR. The ACTIVE trial showed effects of speed of processing training on several cognitive domains and depressive symptoms among older adults. Also, there is published work using computerized cognitive training targeting processing speed on older adults with MCI (for example, see Feng (Vankee) Lin). Additionally, the majority of the work utilizing CR for depression focuses on cognitive control, a component related to executive function. Thus, it would be beneficial if the authors provide an alternative conclusion for these mixed findings that it is specific for patients with MCI.
Thank you very much for this hint. We have added the results of a Cochrane review by Gates et al. who also reports the lowest effects for processing speed and EF. However, we also have mentioned the results of Feng et al., 2016 which found effects on processing speed and was not considered (for whatever reason) in the two meta-analyses.
To our knowledge, the ACTIVE trial was not designed for people with MCI but for older adults with an MMSE - score >= 23. And we have deleted the sentence saying that processing speed and executive function are rarely targeted by CR ?.
Line 176: the phrase “in the same way” is repeated twice.
Oh, thank you very much. We have deleted one phrase.
Line 180: I believe the authors meant to say “studies” rather than “stories”
You are absolutely right, thank you. Please excuse our negligence.
Line 181: correct to say “patients with dementia” rather than “people with dementia patients”
To be honest, we wanted to use the term “people with dementia”. Please excuse, we have corrected that.
Line 188: This is the first time the authors mention “CT”, I assume they are referring to cognitive training? It would help to spell it out given it is the first time they mention “CT”, or simply use CR to be consistent throughout the manuscript.
Oh yes, thank you so much. Fortunately, this is the only time we have erroneously used the term CT in our manuscript.
Line 234: It would be good for the authors to describe why and how they assume that CR would have beneficial effects on social competence, self-efficacy, and sense of mastery. This is a relatively novel idea and should have some discussion and literature to support it.
Thank you very much. We were not aware that this idea is a novel one. We have now explained how such beneficial effects might arise in more detail.
Line 280: the paragraph suggesting metacognition as a combinable approach to CR appears to be underdeveloped. It would be helpful for the authors to elaborate on how metacognition can be applied to develop and use strategies during training and how applying metacognition can facilitate transfer of the effects of training on neurocognition to functional outcomes.
Very good idea, thanks a lot. We have expanded on this topic now.
Section 3.5: in this section, the authors make a strong case for intervening to improve social cognition. However, the argument falls week for responding to why this is important. As it reads, the argument appears to support targeting social cognition on those with difficulties with social interactions or interpersonal skills, such as individuals with Schizophrenia. Although the authors do mention that social anxiety may be a common thread with other disorders, it is not entirely clear how a combined intervention targeting social cognition and neurocognition can benefit an array of individuals with psychiatric disorders and cognitive deficits. Thus, the authors should expand on this.
Oh, thank you. Our chain of arguments was that social cognition is important for social outcome and that besides training in social cognition (for which the evidence is mixed), the social components of CR could be strengthened in a targeted manner to increase its effects. We have added some additional literature about the importance of social cognition and have slightly changed the flow of this section. We hope it is easier for the readers now to understand our thoughts.
Section 3.6: on the section “Help to build bridges” the authors provide excellent concrete examples on the application of cognitive training to everyday functional activities and how these can facilitate better training outcomes.
Thank you very much for the compliment. Writing this part of the manuscript was also a lot of fun.
Section 3.7: regarding conjectures about the effects of CR on non-cognitive domains in MDD and schizophrenia, Quiñones et al. 2020, which the authors already cite, propose that CR may strengthen functional connectivity of functional regulatory areas associated with emotion regulation and thus this may be worth mentioning as another potential explanation for CR effects on clinical symptoms of these disorders.
Great point. We have added these thoughts.
Section 3.8: it would be helpful for a broader audience for authors to define the concept of “hot cognition.”
Thank you very much. Hot cognition tasks use emotionally relevant stimuli instead of stimuli without any emotional influence. We have added this definition.
Section 3.9.1 – part 1: the statement that CR often takes place in groups may be an overgeneralization from CR interventions with schizophrenia. This is the same issue as with section 3.5. CR in studies with other patient populations such as patients with depression, PTSD, or MCI is commonly delivered individually, and this is also true for clinicians delivering CR in clinical settings.
You are right. We have worded this section more carefully now and encourage clinicians to try CR group sessions as well.
Section 3.9.1 – part 2:
The authors should state clearly how social cognition exercises benefit individuals with difficulties in social interactions and how this approach could further benefit individuals with other mental health disorders.
Again, Thank you very much. This is true. Following your recommendations, we have explained in more detail, how social cognition exercises could help in section 3.2 and have extended section 3.5. In this section, we refer to the explanations in these chapters. We hope it becomes more apparent now why we advocate for social cognition training that is naturally embedded in the social context of traditional CR inerventions.
Section 3.9.2: the paragraph on physical exercises and CR appears to be inconclusive. It would be helpful if the authors addressed how combined physical exercises and CR may improve cognitive and clinical symptoms. For example, exercise promotes cognitive and emotional brain pathways that are also strengthened by CR, resulting in improved cognitive function and clinical symptoms of psychiatric disorders.
You are right, thank you very much. We have added some more references and reported current hypotheses regarding putative improvement mechanisms.
Throughout the manuscript:
It is very important that authors avoid using biased-language. For example, the field of behavioral sciences has moved from using the word “suffer” to other descriptors such as “experience.” Another example in the manuscript that the authors should avoid is “psychiatric patients”; describe individuals as “patients with psychiatric disorders” or “patients with ADHD.”
Thank you. We apologize for this and have changed our manuscript accordingly.
The flow of section 3 does not follow the flow of Figure 1. Consider structuring this section according to the figure to facilitate the flow or timing of the recommendations.
Thank you for this very helpful hint. We have changed the flow of section 3 so that it now matches Figure 1.
Authors are encouraged to revise the manuscript for English language grammatical structure.
The English of the manuscript has been heavily revised, and we are confident that it now contains fewer language errors.
Reviewer 2 Report
This article is a review (not systematic) on cognitive remediation (CR) in psychiatric disorders, including not only schizophrenia and other psychoses (diseases widely studied from cognitive rehabilitation) but also others such as ADHD, PTSD, substance abuse or eating disorders. The paper starts by reviewing the cognitive deficits in these disorders, their relevance to functioning in daily life and the evidence for the effectiveness of CR in each of these disorders. The authors then highlight the factors of CR that they consider most relevant and according to the evidence in the literature.
In my opinion, this is an original, up-to-date and relevant work for the application of CR in the clinical context. The paper is well written, and the scientific evidence is ample, correctly referenced, current and relevant to each section. I enjoyed reading the review, congratulate the authors for their good work , here has some review comments:
- I think the issue of "future perspectives" (as the title states) is not fully clarified in the paper. Although the authors make their proposal of what a more efficient CR should be, I miss more of a perspective or emphasis on what they believe these techniques will or should look like in the near future.
- Section 1.1. For example, “There is robust evidence regarding stable deficits in a wide range of cognitive domains, such as attention, verbal and visual (working-) memory, and executive functions of about one standard deviation below average.” It is true that there is ample evidence of cognitive impairments in schizophrenia and other psychoses. However, I believe that more emphasis needs to be placed on the wide heterogeneity of psychoses in their clinical manifestation, which includes cognitive deficits (in addition to many other candidate biomarkers or endophenotypes).
- Section 2. “However, when CR is combined with adjunctive psychiatric rehabilitation and when social functioning instead of cognitive functioning is considered, strategy-based training approaches appear to be superior to pure ‘drill and practice’ approaches.” Does psychiatric rehabilitation refer to pharmacological treatment, psychotherapy or both?
- Section 2. Please indicate the full name of the RCT the first time it is mentioned (line 133). Verify that subsequent times it is mentioned with the acronym and not the full name (line 146). Also check for the acronym CR throughout the text.
- Section 2. “[…] people with dementia patients […]” I think this is a typo and should be either “people with dementia” or “dementia patients”.
- Section 3. Sections 3.1 to 3.10 correspond to the factors that the authors address and suggest as effective CR strategies. Why does the order of these sections not correspond to the order "before, during, after” of CR (e.g., Figure 1)? What is the reason for the different order of these sections?
- Section 3.2 “only patients that suffer from significant cognitive impairments” What aspects or factors would mark the threshold of "significant cognitive impairment". I believe this issue should be briefly addressed.
- Section 3.7 I believe that this section should delve somewhat deeper into the specific beneficial effect of pharmaceutical intervention on clinical symptomatology, which secondarily may have an impact on cognitive performance. In my opinion a greater focus on the pharmacological issue is required in the review (although this is not its main subject) and this section could be its place.
- Section 3.9.1 “One reason, why the combination of rehabilitation and CR is superior to CR alone is the fact that participating in rehabilitation is hardly possible without having a social exchange with other participants or without using the cognitive skills acquired by the cognitive remediation intervention” What do you mean by "rehabilitation" differentially to RC?
- Figure 3. There is a series of colours (blue, green, orange...) that escapes me if they have any specific meaning. Does it have any relation with the colours in figure 1?
Author Response
Thank you very much for your thorough review, encouraging words, and many helpful comments. We think that the quality of our manuscript has improved considerably after considering your hints.
- I think the issue of "future perspectives" (as the title states) is not fully clarified in the paper. Although the authors make their proposal of what a more efficient CR should be, I miss more of a perspective or emphasis on what they believe these techniques will or should look like in the near future.
Thank you very much for this hint. We thought that especially chapters 3.6. (bridges), 3.8 (hot c.) and 3.9 (o.o.t.B.) presented some ideas of what the future of CR could look like. Nevertheless, you are right, and the manuscript would benefit from a separate section addressing our vision regarding the future of CR. We have placed these thoughts in our manuscript's “conclusions” chapter.
- Section 1.1. For example, “There is robust evidence regarding stable deficits in a wide range of cognitive domains, such as attention, verbal and visual (working-) memory, and executive functions of about one standard deviation below average.” It is true that there is ample evidence of cognitive impairments in schizophrenia and other psychoses. However, I believe that more emphasis needs to be placed on the wide heterogeneity of psychoses in their clinical manifestation, which includes cognitive deficits (in addition to many other candidate biomarkers or endophenotypes).
Thank you so much, this is a very good point. We have addressed this issue now.
- Section 2. “However, when CR is combined with adjunctive psychiatric rehabilitation and when social functioning instead of cognitive functioning is considered, strategy-based training approaches appear to be superior to pure ‘drill and practice’ approaches.” Does psychiatric rehabilitation refer to pharmacological treatment, psychotherapy or both?
Thank you for this important hint. The term is not really well defined, but non-pharmacological interventions like social skills training, vocational rehabilitation or supported employment are usually meant. We have added this information.
- Section 2. Please indicate the full name of the RCT the first time it is mentioned (line 133). Verify that subsequent times it is mentioned with the acronym and not the full name (line 146). Also check for the acronym CR throughout the text.
Done, thank you very much.
- Section 2. “[…] people with dementia patients […]” I think this is a typo and should be either “people with dementia” or “dementia patients”.
You are right, we have changed the text to “people with dementia”.
- Section 3. Sections 3.1 to 3.10 correspond to the factors that the authors address and suggest as effective CR strategies. Why does the order of these sections not correspond to the order "before, during, after” of CR (e.g., Figure 1)? What is the reason for the different order of these sections?
To be honest, I made the graph at the very end. But you are absolutely right, the order of the sections should correspond to the order "before, during, after” in figure 1 and we have changed this now. Thank you so much for this comment.
- Section 3.2 “only patients that suffer from significant cognitive impairments” What aspects or factors would mark the threshold of "significant cognitive impairment". I believe this issue should be briefly addressed.
You are absolutely right. Thank you very much. Usually, a score of at least 1 SD below the normative mean is used as a criterion. We have added this information now.
- Section 3.7 I believe that this section should delve somewhat deeper into the specific beneficial effect of pharmaceutical intervention on clinical symptomatology, which secondarily may have an impact on cognitive performance. In my opinion a greater focus on the pharmacological issue is required in the review (although this is not its main subject) and this section could be its place.
This is a very good idea. Although we are no experts in pharmacotherapy and cognition, we have added a few lines pointing out that little attention has been drawn to the impact of different drugs on the effects of CR. As an example, we have cited Nuechterlein’s findings concerning the impact of oral vs. long-acting injectable medication on the effects of CR.
- Section 3.9.1 “One reason, why the combination of rehabilitation and CR is superior to CR alone is the fact that participating in rehabilitation is hardly possible without having a social exchange with other participants or without using the cognitive skills acquired by the cognitive remediation intervention” What do you mean by "rehabilitation" differentially to RC?
We referred to psychiatric rehabilitation interventions described in section 2, and we have now expressed this more clearly in the text
- Figure 3. There is a series of colours (blue, green, orange...) that escapes me if they have any specific meaning. Does it have any relation with the colours in figure 1?
Oh, thanks a lot. In fact, I used different colors to organize the variables a bit when I created the graphic. Actually, I should have made the graphic one color at the end. Instead, I have now added a legend for the colors. What do you think? From your point of view, would the graphic be better in one color or multiple colors with a legend?
Round 2
Reviewer 1 Report
I have no further comments. Thank you for your work.